# Case Report: MRI, CEUS, and CT Imaging Features of Metanephric Adenoma with Histopathological Correlation and Literature Review

**DOI:** 10.3390/diagnostics12092071

**Published:** 2022-08-26

**Authors:** Georg Gohla, Malte N. Bongers, Sascha Kaufmann, Mareen S. Kraus

**Affiliations:** Department of Diagnostic and Interventional Radiology, University-Hospital of Tuebingen, 72076 Tuebingen, Germany

**Keywords:** metanephric adenoma of the kidney, magnetic resonance imaging, contrast-enhanced ultrasound sonography (CEUS), computed tomography, nephron-sparing surgery (NSS), spiking fever

## Abstract

The metanephric adenoma is an extremely rare, benign, embryonal-epithelial neoplasm of the kidney and has a good prognosis with appropriate treatment. It can present at any age and is often asymptomatic. Histologically, the lesion is well established; however, there have been only a few cases described with available detailed imaging findings, most of them with large renal masses typically depicted by computed tomography (CT). This case report includes imaging of contrast-enhanced MRI, contrast-enhanced ultrasound (CEUS), and CT, and thus adds to the information available, potentially promoting a nephron-sparing clinical pathway. We report on the clinical presentation, imaging, histopathological diagnosis, and treatment data of a 27-year-old female, in whom an incidental, symptomatic kidney tumor was detected. CT, CEUS, and MRI showed a suspicious unifocal renal lesion with inhomogeneous enhancement, which was indistinguishable from renal cell carcinoma. After laparoscopic resection, a metanephric adenoma with microscopically partially glandular, partially nest-like solid growth and without distinctive atypia was diagnosed pathohistologically. Immunohistochemistry results were positive for Wilms Tumor 1 and CD57 and negative for EMA and CK7: 2–3% positive cells in MIB1 coloring. At 3-month and 1-year follow-up, the patient was asymptomatic and imaging showed no recurrence of renal masses or metastases.

## 1. Introduction

The metanephric adenoma (MA) is a rare embryonal epithelial tumor of the kidney. It was first described in 1979 by Bove [1,2] and shows characteristic histology [3]; however, pre-operatively distinguishing MA from renal oncocytoma or malignant tumors including Wilms’ tumor or renal cell carcinoma remains challenging. As epidemiology and imaging presentation of these can be quite similar, and because of its rarity accounting for only 0.2–1% of benign adult renal epithelial neoplasms [4,5], MA is often not well recognized by clinicians and radiologists. To date, less than 200 cases of MA have been reported on in English language literature. Of those, only a few have been well documented with morphological and imaging features. Currently, MA is considered a benign renal neoplasm; however, a few cases of metastatic behaviorism have been reported [6,7,8]. Most MA cases undergo nephrectomy because of a lack of differentiating imaging from papillary renal cell carcinoma.

We report a case of histologically confirmed MA in a 27-year-old female and describe its appearance on MRI (magnetic resonance imaging), CEUS (contrast-enhanced ultrasound) and CT (computed tomography). As seen in this case and according to the experiences of others in the literature, it is very challenging to make a definite diagnosis based on imaging, and therefore, pathohistological correlation by nephron-sparing surgery is needed.

## 2. Case Study

A 27-year-old woman presented to our regional tertiary hospital due to spiking fevers of unknown origin and occurring every 8 weeks over the past 12 years. At primary presentation in 2012, there were no features of vasculitis, collagenases, or mutation of the MEFV-Gene (Familial Mediterranean Fever). Initial abdominal sonography and laboratory tests were normal. Apart from a known hypothyroidism and a previous inguinal hernia repair, the patient was healthy. Her family history was unremarkable.

The patient presented again in late 2019 in the rheumatology department of our university hospital. The periods of spiking fevers had shortened in duration in the past 5 years, from 7 to 3 days, and become less severe. Often, they occurred with a sore throat, body aches, or abdominal pain. The patient was also a healthcare professional and noted a concordance with high stress levels.

After a thorough but unremarkable clinical examination, blood laboratory was ordered, which showed a slightly low hemoglobin level of 11.2 g/d, lowered erythrocytes with 3.6 Mio/μL, and slightly elevated lymphocytes with 47.3% with at the same time slightly decreased absolute neutrophil counts of 1.8 thousand/μL. Otherwise the results were normal: electrolytes as well as renal specific laboratory parameters were normal (Creatinine at 0.5 mg/dL and GFR < 90 mL/min/1.73 m^2^). Rheumatologic laboratory was non-conclusive. The patient did not have an elevated temperature on the day of presentation and showed to be in a good general condition without hematuria, frequent urination, odynuria, and dysuria.

With the history of these persisting symptoms and normal laboratory, a contrast-enhanced CT examination was ordered for neoplasm screening (Siemens Definition Flash CT acquired 90 s after contrast injection via the cubital vein). In early December 2019, a solid mass was therewith found in the cortex and parenchyma of the lateral midportion of the left kidney. The lesion was well circumscribed, homogeneous hypodense in comparison to the contrast-enhanced parenchyma and measured approximately 12 mm × 11 mm × 11 mm. The density was approximately 90 Hounsfield Units (HU). For imaging details, please see Figure 1. Apart from the incidental finding at the left kidney, there were no other pathologic findings. To further characterize the lesion, a contrast-enhanced ultrasound scan (CEUS) was recommended by our department.

The patient presented again a week later at our department for the CEUS (Acuson S3000 with 6C1 HD curved array probe and 9L4 linear probe, Siemens, Erlangen, Germany). The lesion in the left kidney was still detectable with well-circumscribed borders, a maximum diameter of 12 mm, and was slightly inhomogeneous with a low echo intensity in comparison to the surrounding tissue, see Figure 2A. In color flow Doppler, the lesion showed no significant vascularity (Figure 2B). After contrast injection, there was a slight contrast enhancement notable in the late phase (Figure 2C). A contrast-enhanced MRI was advised for further lesion characterization.

In the following week, the patient was assessed in supine position on a 1.5T MRI scanner (Siemens, Magnetom Aera) with a 64-channel body array coil and a protocol consisting of T1w with additional in- and opposed phase images, T2w, diffusion-weighted and Gadobutrol enhanced T1w imaging in arterial, portal venous, early and late venous phases.

In keeping with the previous imaging, the renal mass was well-demarcated without a fibrous capsule with a slight bulge of the intact renal outline. In the T2-weighted sequence the lesion showed a hypointense signal in relation to the surrounding renal cortex and medulla (Figure 3A). In the non-contrast-enhanced T1-weighted images, the lesion had a heterogeneous slightly hypointense signal in comparison to the surrounding renal cortex (Figure 3B). In diffusion-weighted images, the lesion showed a slight diffusion restriction (Figure 3C,D). Additional T1-weighted in- and opposed-phase sequences revealed no further information; there was no evidence of fatty tissue or iron deposition.

After i.v. injection of 6 mL Gadobutrol (Gadovist^®^, Bayer, Leverkusen, Germany) contrast-enhanced fat saturated T1-weighted images showed a slight heterogeneous enhancement of the lesion, especially in portal venous and venous phases, with an indicated wash-out kinetic in late venous phase (Figure 3E–H). Consistently, the lesion was hypointense in all contrast-enhanced images in relation to the renal cortex.

The benign characteristics of the lesion were the well-defined circumference and a renal border bulge. However, there was a slight diffusion restriction accompanied by contrast media uptake with an indicative washout kinetic compared to renal cortex. Therefore, a malignant underlying condition, such as papillary renal cell carcinoma (PRCC), could not be excluded.

Consequently, the patient received a robot-assisted (Davinci) laparoscopic partial nephrectomy of the left kidney in March 2020. Intraoperative sonography was conducted, and the intraoperative specimen revealed a slight buckling of the tumor at the renal cortex and a faintly brown coloring. The lesion did not show a communication to the collective system and was removed without complication. At the first follow-up 3 months after surgery, ultrasound showed normal postoperative findings. At 12-month follow up, the patient was clinically symptom-free (without the undulant fever) and in non-enhanced MRI there was no renal recurrence detectable (Figure 4).

The macroscopic findings of the pathohistological report were a completely imbedded, brownish tumor, which measured 17 mm × 6 mm × 6 mm. Microscopically, the tumor cells were partially glandular, partially nest-like with solid growth and without distinctive atypia. The tumor cells were middle-to-large with a narrow cytoplasm and a large cell nucleus. There was chromatin homogeny and overall low pleomorphia. Immunohistochemistry results were positive for Wilms Tumor 1 and cytoplasmic CD57 and negative for epithelial membrane antigen (EMA) and CK7; 2–3% positive cells in MIB1 coloring. The pathological diagnosis was MA of the left kidney without an indication for malignancy.

## 3. Discussion

The metanephric adenoma is a rare epithelial tumor derived from metanephrogenic embryonic tissue, which can occur at any age but has a diagnosis peak in the fifth or sixth decade of life and a female predominance of 2:1 [8,9,10]. It features mainly characteristics of a benign tumor with a low proliferation rate and favorable outcome. Due to the tumor’s rarity, it is not typically on the list of differential diagnoses of most clinicians, radiologists, or surgeons. Patients with MA are usually asymptomatic, and the diagnosis is made incidentally by routine investigations. Some, however, present with non-specific abdominal symptoms such as abdominal or flank pain, painless hematuria, intermittent fever and, according to some authors, polycythemia; MA cells secrete erythropoietin and multiple cytokines [10,11]. Occasionally, a palpable tumor in the upper abdomen has been seen. MA is closely related to the Wilms’ tumor–some authors even say that MA is the benign counterpart [9], but it has also overlapping features with the PRCC, especially in imaging features. [3]. However, there are case reports of metastatic tumor spread to lymph nodes [12,13] or malignant cells [14,15]. There are no typical locations in kidneys in which MA usually arises, but the cortex is often affected and typically it is a unilateral growth but there have been singular cases with MA arising bilaterally [16].

On imaging, MA presents as a well-defined, round or oval, soft tissue lesion with often bulging of the outer cortex contour. Most MA show to be non-encapsulated, conversely most PRCC are bordered by a thick fibrous pseudo-capsule [3]. On ultrasound, MA shows non-specific characteristics ranging from a low- to iso- to high-echogenic solid mass without significant vascularity on color flow Doppler imaging [9,17]. Our patient presented with a well-defined, approximately 1.2 cm lesion with inhomogeneous, low echogenicity in the midportion of the left kidney and typically with no significant flow void in color Doppler imaging. However, after contrast administration on ultrasound, a slight increase in blood flow was noted after 3 min.

CT typically reveals a well-defined, non-invasive, solid, and mostly unilateral and unifocal lesion, which can include calcifications (20%), cystic spaces and necrosis [17,18]. MA varies greatly in size and mostly presents with an intact renal border [8,9].

In this case, MA appeared as a well-demarcated soft tissue mass with contrast sparing compared to the enhanced adjacent renal parenchyma/cortex. There were no calcifications or cystic transformations apparent. Contrast media uptake could not be interpreted due to the lack of unenhanced imaging or dual energy application to reconstruct virtual monoenergetic reconstructions.

There are no distinct attenuation patterns described for MA in CT. According to the literature, MA can be depicted as a slightly hyper- or hypodense lesion compared to the underlying parenchyma and can show enhancement in dynamic CT examinations. However, the enhancement degree is lower than the renal cortex in both the corticomedullary and nephrogenic phase with, at times, a gradual and prolonged enhancement ranging from primarily peripheral to centripetal enhancement [19,20,21].

Such non-specific characteristics are also seen in MRI; however, this imaging technique often cannot further elucidate the diagnosis. MA has been previously described as a hypointense mass in T1 weighted imaging with iso- or slightly hyperintense imaging characteristics on T2 weighted imaging [8,22]. In keeping with this, our case was shown to have a well-demarcated renal mass without a fibrous capsule with slight hypointensity in the T1-weighted imaging. However, in the T2-weighted images the lesion was shown to have a hypointense signal, in relation to the surrounding renal cortex and medulla, with a slight restriction on diffusion-weighted images. Contrast-enhanced T1w images revealed a slight inhomogeneous mass with a prolonged contrast media uptake lower than the renal cortex. In late venous phase, the tumor showed an indicated wash-out compared to renal cortex.

Imaging is inadequate to differentiate MA from other renal masses because specific imaging characteristics do not exist. Pathological diagnosis is definitive but pre-operative biopsy uncommon.

Macroscopically MA is described as a solitary, well defined tumor with varying size, coloring (from grey to tan to yellow) and tissue stiffness (soft or firm) [8]. As discussed, cystic, hemorrhagic, or necrotic transformation has been described and correlates to the tumor variability on T1 and T2 weighted images in MRI.

Microscopically, MA is composed of tightly packed small and uniform (round to oval) cells, with scanty cytoplasm, arranged in solid sheets or small acini-like aggregates, minimal intercellular stroma with various degrees of intra-tumoral fibrous bands, absence of mitoses and intermingled with glomerular or papillary architectural patterns [1,3]. Psammoma bodies have also been described in MA with possible dystrophic calcification and macrophages [3].

Additionally, immuno-histochemical characterization usually shows an expression of CD57, often CK7 cytoplasmic immunoreactivity (according to Padilha et al. in 57% of the cases and especially in areas of tubular formation) and WT1 without showing epithelial membrane antigen (EMA) or AMACR [3,19,22,23]. These features are essential in differential diagnosis of the solid variant of papillary renal cell carcinoma (s-PRCC), which presents strongly positive for CK7 and AMACR with epithelial membrane antigen reactivity, whilst being negative for WT1 [3,6]. Additionally, s-PRCC may show micronodules of tumor cells or tubular structures in contrast to the tight small tubules, acini, or solid areas seen in MA [6]. In our case, the immunohistochemistry results were positive for WT1 and cytoplasmic CD57 and negative for epithelial membrane antigen (EMA) and CK7.

Genetic evidence suggests that metanephric adenoma is its own entity and can be differentiated from Wilms’s tumor, which shows an allelic imbalance at chromosome 11p13 [24]. There is inconsistent evidence regarding genetic similarities between MA and PRCC: Brown et al. reported a frequent chromosome 7 and 17 gain and sex chromosome loss in both entities, this, however, could not be confirmed by others who instead found cytogenetically normal diploid karyotypes [8,25,26,27]. Ding et al. recently suggested that novel clinicopathological and molecular features (dual specificity phosphatase family of protein 6-induced extracellular signal-regulated kinase ½ dephosphorylation) could provide benefits for the diagnosis and better understanding of MA [28].

## 4. Conclusions

This case has presented comprehensive radiological characteristics of MA including CEUS, CT, and MRI features with correlation to histopathology. It was depicted as a well-circumscribed, heterogeneous renal mass protruding the renal outline with a slight diffusion restriction and a subtle contrast media uptake, which however, was less than the underlying parenchyma/cortex.

MA should be considered in the differential diagnosis of malignant renal tumors, even though it is a rare entity. Since MA does not have specific clinical symptoms nor clear imaging characteristics that allow for unambiguous diagnosis, a differentiation of MA and other renal masses remains challenging. Due to the lack of differentiation from RCC, operation or biopsy should be performed. Likewise, patient management should not differ from that of RCC until histological confirmation of MA. However, isolated metastatic spread of MA has been reported and a consensus in follow-up imaging recommendation would be desirable.

## Figures and Tables

**Figure 1 diagnostics-12-02071-f001:**
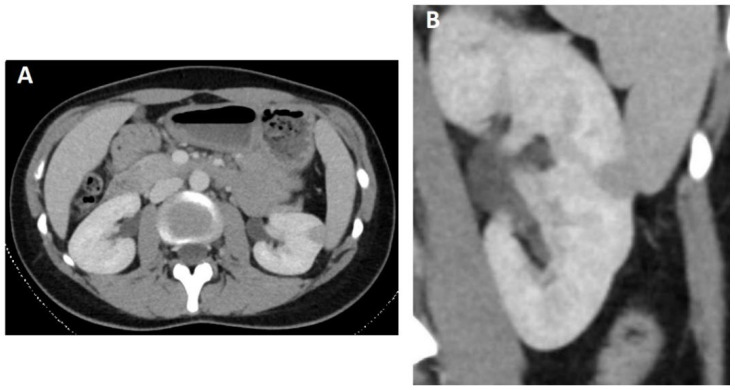
Enhanced CT in axial (**A**) and coronal (**B**) plane show a well-defined bulging mass in the midportion of the left kidney with an almost homogeneous and hypodense characteristic compared to the renal cortex in portal venous phase.

**Figure 2 diagnostics-12-02071-f002:**
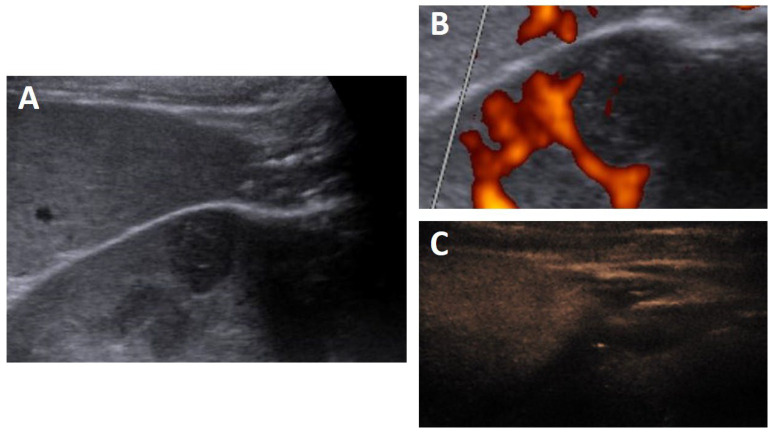
Ultrasonography demonstrates a hypoechoic lesion with small linear slight hyperechoic lines (**A**) without significant blood flow in color flow Doppler (**B**) but with a slight contrast enhancement in the late phase of contrast-enhanced ultrasound (CEUS) (**C**).

**Figure 3 diagnostics-12-02071-f003:**
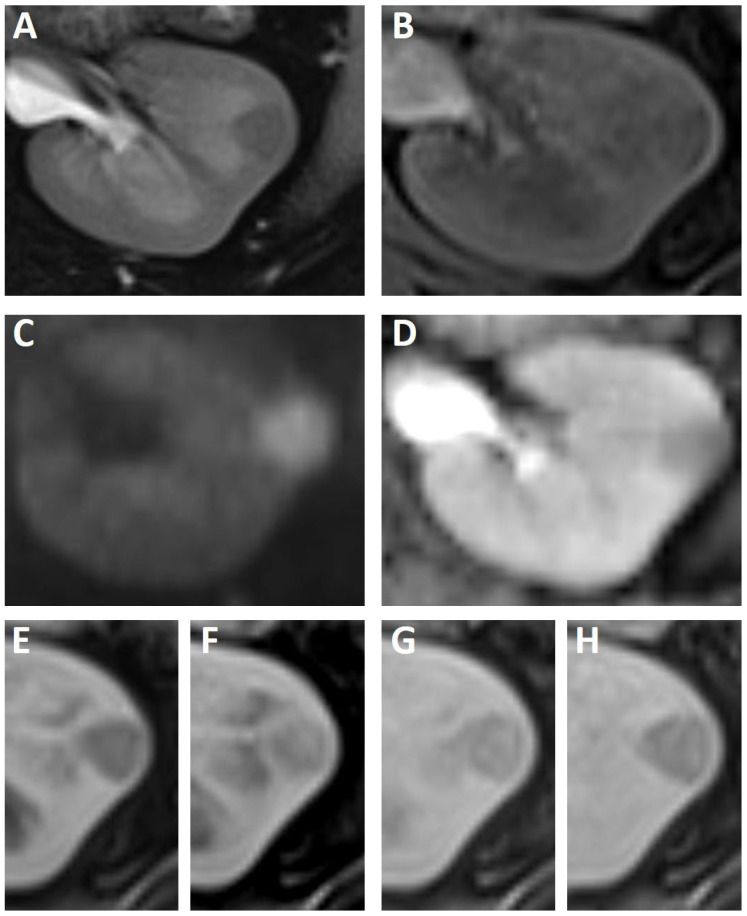
MRI findings of the left kidney tumor in axial planes. In this case, the bulging metanephric adenoma is with hypointense signal in T2-weighted (**A**) as well T1- weighted (**B**) images. In b800 s/mm^2^ diffusion-weighted sequences the lesion is hyperintense (**C**) with a corresponding low signal in the apparent diffusion coefficient map (**D**). Contrast-enhanced images in arterial (**E**), portal venous (**F**), early venous (**G**) and late venous (**H**) phase demonstrate a subtle delayed enhancement of the renal mass with an indicated slight wash out in late venous phase. However, the lesion is in all imaging hypovascular compared to the renal cortex.

**Figure 4 diagnostics-12-02071-f004:**
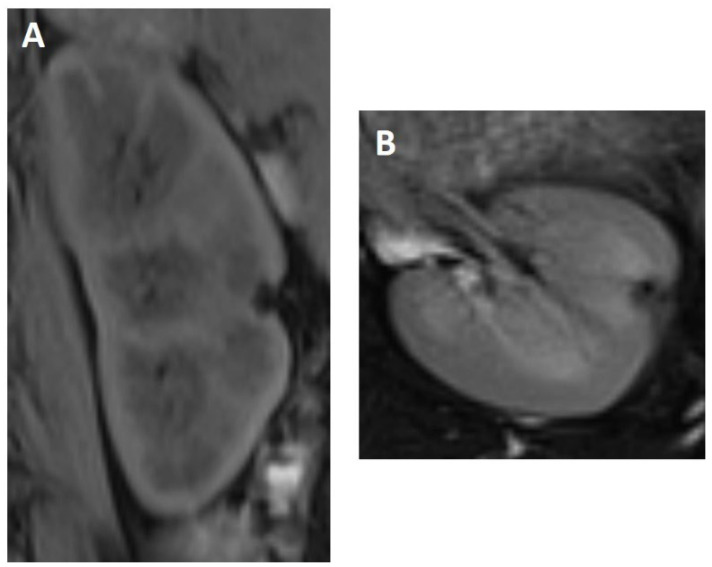
Postoperative 1-year follow-up shows a small parenchymal defect in the midline of the left kidney without evidence of recurrence (T1 coronal with fatsat (**A**) and T2 axial with fatsat (**B**)).

## Data Availability

Not applicable.

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
