# Peer review of "Case Report: MRI, CEUS, and CT Imaging Features of Metanephric Adenoma with Histopathological Correlation and Literature Review"

_diagnostics, 2022, doi:10.3390/diagnostics12092071_

Round 1
Reviewer 1 Report
final sentence in the abstract (with appropriate treatment, it has a good prognosis.) should be moved to the beginning/introduction
The case is well presented with good figures showing the imaging characteristics of the MA
Authors claim that MA should be considered as a differential diagnosis in renal lesions in order to improve patient management. However, the lesion shows similar imaging features to malignant lesions. Since MA is very rare compared to RCC, patient management shouldn't be changed because of this rare DD. Surgical resection is recommended when RCC is suspected. This should become more clear in the discussion and in the abstract.
Author Response
Thank you for the detailed review and the useful comments. We have incorporated the suggestions you have made in the revised manuscript accordingly.
Reviewer 2 Report
You report on a case of Metanephric adenoma and review the diagnostic imaging and the histopathology of the lesion. The first case report was by Bove in 1979 (Spaner SJ Pediatric metanephric adenoma: case report and review... in Int UroL Nephrol 2014;46, 677-80). Did you have a preop UA? Do you see any lymph node, any fat, any calcification at any time? Do you see any mitotic activity, any psammoma on the section?
Thank you for the privilege of reviewing the manuscript.
Delete This on line 232
Author Response
Thank you for your review and the helpful comments. As suggested, we have corrected the original description of MA and deleted the typo in line 232 – thank you for pointing that out. Our colleagues performed a preoperative urinalysis one day before surgery, which was normal and showed a pH of 6.0. We did not see any fat or calcifications within the lesion or any notably enlarged lymph nodes. Histologically, the section showed no mitotic activity. Unfortunately, information about psammoma is not available to us.